# Screening of Anorectal and Oropharyngeal Samples Fails to Detect Bacteriophages Infecting *Neisseria gonorrhoeae*

**DOI:** 10.3390/antibiotics11020268

**Published:** 2022-02-18

**Authors:** Jolein Gyonne Elise Laumen, Saïd Abdellati, Sheeba Santhini Manoharan-Basil, Christophe Van Dijck, Dorien Van den Bossche, Irith De Baetselier, Tessa de Block, Surbhi Malhotra-Kumar, Patrick Soentjes, Jean-Paul Pirnay, Chris Kenyon, Maia Merabishvili

**Affiliations:** 1Department of Clinical Sciences, Institute of Tropical Medicine, 2000 Antwerp, Belgium; sabdellati@itg.be (S.A.); sbasil@itg.be (S.S.M.-B.); cvandijck@itg.be (C.V.D.); dvandenbossche@itg.be (D.V.d.B.); idebaetselier@itg.be (I.D.B.); tdeblock@itg.be (T.d.B.); psoentjes@itg.be (P.S.); ckenyon@itg.be (C.K.); 2Laboratory of Medical Microbiology, Vaccine and Infectious Disease Institute, University of Antwerp, Campus Drie Eiken, 2610 Wilrijk, Belgium; surbhi.malhotra@uantwerpen.be; 3Center for Infectious Diseases, Queen Astrid Military Hospital, Neder-over-Heembeek, 1120 Brussels, Belgium; 4Laboratory for Molecular and Cellular Technology (LabMCT), Queen Astrid Military Hospital, Neder-over-Heembeek, 1120 Brussels, Belgium; jean-paul.pirnay@mil.be (J.-P.P.); maya.merabishvili@gmail.com (M.M.); 5Department of Medicine, University of Cape Town, Cape Town 7701, South Africa; 6Microbiology and Virology (EIBMV), Eliava Institute of Bacteriophage, Tbilisi 0162, Georgia

**Keywords:** *Neisseria gonorrhoeae*, bacteriophage screening, antimicrobial resistance, phage therapy, human anorectal swabs, human oropharyngeal swabs

## Abstract

There are real concerns that *Neisseria gonorrhoeae* may become untreatable in the near future due to the rapid emergence of antimicrobial resistance. Alternative therapies are thus urgently required. Bacteriophages active against *N. gonorrhoeae* could play an important role as an antibiotic-sparing therapy. To the best of our knowledge, no bacteriophages active against *N. gonorrhoeae* have ever been found. The aim of this study was to screen for bacteriophages able to lyse *N. gonorrhoeae* in oropharyngeal and anorectal swabs of 74 men who have sex with men attending a sexual health clinic in Antwerp, Belgium. We screened 210 swabs but were unable to identify an anti-gonococcal bacteriophage. This is the first report of a pilot screening that systematically searched for anti-gonococcal phages directly from clinical swabs. Further studies may consider screening for phages at other anatomical sites (e.g., stool samples, urine) or in environmental settings (e.g., toilet sewage water of sex clubs or sexually transmitted infection clinics) where *N. gonorrhoeae* can be found.

## 1. Introduction

Bacteriophages (or phages) are viruses that interact with and invade bacterial cells resulting in the lysis of the infected host (virulent phage) or the integration of the phage genome in the chromosome of the host cell (temperate phage). With an estimated 10^31^–10^32^ phages in total, they are the most abundant living entities on Earth and play a crucial role in the microbial balance of the planet [1]. In recent years there has been renewed interest in the use of phages to prevent and contain the emergence of antibiotic resistance [2,3,4,5].

*Neisseria gonorrhoeae*, the bacterial pathogen causing the sexually transmittable infection gonorrhea, has developed resistance to all previously and currently recommended antimicrobials. As such, there are valid concerns it may become untreatable with conventional antimicrobials [6,7]. Antimicrobial resistance in *N. gonorrhoeae* is driven by excessive antimicrobial exposure [6,8]. Phages active against *N. gonorrhoeae* could play an important role in preventing the emergence of untreatable gonorrhea in two ways. Firstly, as an antibiotic-sparing therapy, which could reduce the selection pressure for the emergence of resistance against conventional antimicrobials. Secondly, they could be used in combination with antimicrobial agents to treat multi-resistant *N. gonorrhoeae* [9,10]. Phages have a number of advantages over antimicrobial therapy. They can be extremely species- and strain-specific and could be chosen not to have a major impact on the commensal flora [11]. In particular, they will not select for antimicrobial resistance (AMR) in commensal bacteria. This property might especially be beneficial for *N. gonorrhoeae* since anti-gonococcal antimicrobial therapy selects for AMR in commensal *Neisseria,* and these resistance determinants can be taken up by the pathogenic *N. gonorrhoeae* via horizontal gene transfer [12,13,14,15].

The first phages reported against Neisseria were found to be active against the nonpathogenic species N. perflava [16,17,18]. Phages active against N. meningitidis were first detected as early as 1967 [19], and more recently, Aljarbou et al. reported the isolation of lytic phages targeting N. meningitidis from human tooth plaques [20,21]. Remarkably, these anti-Neisseria phages showed extremely limited host ranges, being mostly reactive against a few strains within one species [16,17,18,20].

To the best of our knowledge, no phages active against *N. gonorrhoeae* have ever been found. In 1985, Campbell et al. observed plaques on lawns of two different strains of *N. gonorrhoeae*, but no phage particles active against this pathogenic species were finally isolated and identified [22]. Bioinformatic analyses of *N. gonorrhoeae* strain FA1090 have revealed the presence of four integrated filamentous phage genomes (Ngoϕ6–9) [23]. A phagemid derived from *N. gonorrhoeae* filamentous phage Ngoϕ6 showed an ability to infect and effectively produce progeny phagemids in a broad range of Gram-negative bacteria including *Escherichia coli*, *Haemophilus influenzae*, *Neisseria sicca,* and *Pseudomonas* species [24]. In addition, five prophage encoding regions (Ngoϕ1–5) were identified in strain FA1090.

Two of them (Ngoϕ1–2) may be functional, as they appear to encode all genes necessary for lytic growth, and prophage DNA sequences were detected in culture supernatants and visualized by electron microscopy [25]. However, they were unable to propagate and produce plaques on the *N. gonorrhoeae* and nonpathogenic *Neisseria* strains tested, suggesting that each strain is expressing the appropriate lysogenic control genes.

In conclusion, there is scarce evidence that these prophages are competent to produce phage particles, and we could find no evidence of any lytic phages that could be used to develop phage therapy directed towards *N. gonorrhoeae.* Therefore, the aim of the present study was to screen oropharyngeal and anorectal samples for the presence of phages able to infect *N. gonorrhoeae.*

## 2. Results

### 2.1. Isolation and Propagation of Potential Phages

A total of 194 oropharyngeal and 18 anorectal ESwabs™ from 74 MSM were screened for the presence of phages able to infect *N. gonorrhoeae*. Multiple clear zones were detected, ranging from confluent to opaque lysis (Figure 1).

None of the confluent lysis zones could be replicated through further propagation on the appropriate indicator strain. Clear zones pointed towards one pool of ESwabs™ enriched with different mixes of clinical *N. gonorrhoeae* strains and tested at the two different timepoints. Original individual ESwab™ filtrates comprising that pool were tested using the spot assay too. Several of these ESwabs™ inhibited the growth of some *N. gonorrhoeae* indicator strains, with one swab showing lysis zones on all indicator strains. None of these confluent lysis zones could be replicated through further propagation on the appropriate indicator strain.

Attempts were made to optimize the propagation methods by (i) prolonging the incubation time of the cut-out lysis zones, (ii) enriching the cut-out lysis zones with the appropriate indicator strain, and (iii) adding chloroform to assist the bacterial lysis process after incubation of the cut-out lysis zones. None of these attempts resulted in the replication of lysis zones. Enriched samples from the pool of ESwabs™ that showed clear zones were spotted on five other bacterial species (*Pseudomonas aeruginosa*, *Staphylococcus aureus*, *Escherichia coli*, *Acinetobacter baumannii,* and *Enterococcus faecalis*) but did not show any lytic activity.

### 2.2. Control Experiments

No lysis zones were observed after spotting filtrates of bacterial cultures. Pure ESwab™ medium faintly reduced growth on a limited number of bacterial overlays; however, it was not comparable with lysis zones of enriched ESwab™ samples. After centrifugation, filtration, and freezing at −80 °C up to one year period, E. coli phage GEC-3S infectivity decreased only at a minimal expected concentration of 2 log PFU/mL; in particular, from 2.14 log PFU/mL it dropped to 1.76 log PFU/mL after 3 days and to 0.94 log PFU/mL after 357 days of storage.

## 3. Discussion

We undertook an extensive search for lytic phages in oropharyngeal and rectal samples from individuals with a high incidence of *N. gonorrhoeae* by using a variety of methodologies. To the best of our knowledge, this is the first study that has systematically searched for anti-gonococcal phages directly from clinical swabs. Previous studies have detected phages active against a range of bacterial species from the oral cavity [26,27,28]. A number of studies have found phages infecting specific *Neisseria* species (but not *N. gonorrhoeae*) from dental plaque and oro- and naso-pharyngeal samples [17,18,19,20].

In this study, antibacterial activity that resulted in the formation of zones of lysis of *N. gonorrhoeae* was commonly encountered but could never be replicated through further propagation. Several studies have reported similar issues. Lysis zones on *N. gonorrhoeae* cultures were detected when suspensions of dental plaque were plated, but none of them could be propagated, and further transmission electron microscopy failed to reveal the presence of particles resembling phages [29]. A phage infecting a specific *N. perflava* strain showed lysis when throat isolates were spotted on a *N. gonorrhoeae* strain, but plaques were never seen. Ultraviolet treatment did not reduce lytic activity, suggesting lysis was not caused by phage particles but rather by an unknown component in the bacterial lysates [18].

False positive results due to bacterial killing by media components is a well-known limitation of the spot test [11]. However, we did not observe similar lysis zones during the negative control experiments in which we used bacterial culture filtrates and pure ESwab™ medium. This observation suggests that something is released from the samples that is causing the lysis. Several microbial interactions could play a role. It was recently reported that *N. gonorrhoeae* is killed when it takes up DNA with a differential methylation pattern released by various commensal *Neisseria* species such as *N. elongata* [30]. In addition, *N. mucosa* has been reported to secrete secondary metabolites that exhibit antigonococcal activity unrelated to the previously described DNA-mediated killing, although the antimicrobial metabolites have not yet been characterized [31]. Since these commensals are prevalent in the oropharynx in a high proportion of individuals, there is a commensurately high probability that this inhibitory DNA was present in the clinical swabs we used [15,32].

Bacteriocins produced by other bacterial species are another possible source of *N. gonorrhoeae* lysis [33]. A bacteriocin (Viridin B) produced by *Streptococcus mitis*, inhabiting the human mouth, is known to be bactericidal to *N. sicca* [34], and caerin 1 antimicrobial peptides have been shown to inhibit *N. lactamica* [35]. In addition, a randomized controlled trial showed that inoculation with *N. lactamica* inhibits the carriage of *N. meningitidis* in young adults [36]. Although the exact mechanism is unknown, bacteriocins, uptake of toxic DNA, and microbial competition could be responsible for this non-phage lysis.

To test the plausibility of this hypothesis, we subjected some strains of all different commensal *Neisseria* species obtained from the pharyngeal samples used as a potential phage source to an adapted spot test [37]. In addition, we included ATCC strains of other bacterial species that could have been present in the oropharyngeal and rectal samples tested, including *Staphylococcus aureus*, *Streptococcus pneumoniae,* and *E. coli*. None of the *Neisseria* species tested (*N. meningitidis*, *N. subflava*, *N. cinerea*, *N. lactamica*, *N. oralis,* and *N. elongata*) showed any effect. However, *S. pneumoniae*, and to a lesser extent *E. coli*, produced significant inhibition zones on all *N. gonorrhoeae* target strains tested.

Other factors in addition to bacterial products may be causing the *N. gonorrhoeae* inhibition during our phage screening experiments. In a study that detected a novel phage targeting *N. subflava*, single plaques derived from dental plaque swabs were used to re-infect the same host-culture; only the first and second round of propagation resulted in plaques [21]. The exact mechanism of losing infectivity or developing host resistance is unclear but is a common observation [22].

Previously reported autoplaquing in *N. gonorrhoeae* [22] could be explained by autolysis or the presence of two different functionally active prophages [25,38]. However, we did not observe any lysis zones during the negative control experiments in which we used bacterial culture lysates in GBC+.

A limitation of our study is that we mainly screened oropharyngeal samples, a limited number of anorectal swabs, and no urethral, urine, dental plaque, female genital tract, or environmental samples (e.g., toilet wastewater). While commensal *Neisseria* species colonize the oral and nasopharyngeal cavities, the median duration of oropharyngeal *N. gonorrhoeae* infection among MSM has been reported to persist between 28 days and up to 16 weeks when untreated [39,40]. Future studies could aim at isolating phages infecting non-pathogenic *Neisseria* species and then trying to train and adapt its host range to ultimately be capable of infecting *N. gonorrhoeae.* Furthermore, some samples were taken after the use of a daily Listerine mouthwash (54 out of 210 samples) or an antibiotic (72 out of 210 samples) in the three months prior to sampling. These may have altered the microbiota in a way that decreased the probability of detecting relevant phages.

## 4. Materials and Methods

### 4.1. Bacterial Strains

Three mixes, each consisting of three or four clinical *N. gonorrhoeae* strains collected at the Institute of Tropical Medicine in Antwerp, Belgium, between 2019 and 2020, were used for phage enrichment. In addition to the 11 enrichment strains, 9 WHO *N. gonorrhoeae* reference strains were included for phage isolation [41,42]. We refer to this collection as our ‘indicator strains’; for further details see Appendix A.

### 4.2. Potential Phage Containing Samples

Oropharyngeal and anorectal ESwabs™ (COPAN Diagnostics Inc., Brescia, Italy) were collected from men who have sex with men (MSM) as part of two clinical studies conducted at the Institute of Tropical Medicine (ITM) in Antwerp, Belgium.

The first study, Preventing Resistance in Gonorrhoea (PReGO), was a placebo-controlled crossover randomized clinical trial assessing whether the regular use of an essential oil-based mouthwash could reduce the incidence of sexually transmitted infections in 343 MSM taking HIV preexposure prophylaxis at ITM [43] (registered at clinicaltrials.gov as NCT03881007). The second study, the Resistogenicity study, was an observational study that evaluated the effect of ceftriaxone and azithromycin on the antimicrobial susceptibilities of commensal *Neisseria* in 10 MSM attending ITM [32]. For further details see Table 1.

All oropharyngeal samples were taken by rubbing both tonsillar pillars and the posterior oropharyngeal wall by study physicians. Anorectal swabs were generally taken by the study participants themselves. After collection, the ESwabs™ were inoculated on Colombia Agar with 5% Sheep Blood and GC-Lect^TM^ agar plates (Becton Dickinson) to isolate *Neisseria* species as part of the two above-mentioned clinical studies. The following *Neisseria* species were present in samples from the PReGO study: *N. sublfava/flavescens*, *N. macacae/mucosa*, *N. meningitidis,* and *N. gonorrhoeae* (unpublished data).

In addition to the above species, N. oralis was found in samples from the Resistogenicity study [32]. The remaining ESwab™ transport media were stored at −80 °C after transfer into cryotubes. Frozen ESwab™ transport media were thawed and centrifuged at 5000× *g* for 15 min in 0.22 µm filter tubes (Costar™, Spin-X™ Centrifuge Tube Filters CA Membrane) to filter out bacteria, cells, and debris present in the sample. The supernatant was stored at 4 °C and used, within 24 h, for phage enrichment.

### 4.3. Phage Enrichment Culture

After filtration, 450 µL media of each ESwab™ was used to generate pools of approximately 20 samples. Each pool was divided into 3 parts (circa 150 µL media from each ESwab™). Each fraction was incubated in 10 mL gonococcal broth (double-distilled water (ddH_2_O) supplemented with 15 g/L Bacto proteose peptone no. 3, 1 g/L soluble starch, 4 g/L K_2_HPO_4_, 1 g/L KH_2_PO_4_, 5 g/L NaCl, and 10 mL/L IsoVitaleX; further referred to as “GCB+”) and with a different mix of enrichment strains at 37 °C in 5% CO_2_ (using 10 µL of 10^7^ CFU/mL suspension of each strain) (Table 1). The turbidity of the cultures was measured, and a 1 mL sample was taken in the mid-log phase (after 6–7 h). Six milliliters of fresh GCB+ was added to supply sufficient nutrition for overnight incubation, whereafter a second sample was taken (18–21 h). Samples were centrifuged at 5000× *g* for 15 min in 0.22 µm filter tubes (Costar™, Spin-X™ Centrifuge Tube Filters CA Membrane) to remove bacteria, and samples were stored at 4 °C until spot testing was carried out.

### 4.4. Spot Test

The spot test method was used as an initial indicator test to screen enriched samples for the presence of phages by measuring lytic activity. An amount of 3 µL of GCB+ containing 0.6% agar and 100 µL of a 10^8^ CFU/mL *N. gonorrhoeae* indicator strain culture was poured onto a gonococcal base (BD Difco^TM^) supplemented with 1% IsoVitalex. The agar mixture was swirled to produce a uniform top layer. Once solidified, 10 µL of potential phage suspensions, obtained after enrichment, centrifugation, and filtering, was spotted onto the overlay. Plates were incubated at 37 °C in 5% CO_2_ for 24 to 48 h before inspection of cleared zones.

### 4.5. Phage Isolation and Propagation

Confluent lysis zones were cut with a sterile inoculating loop and inoculated into 4 mL GCB+ in 14 mL sterile tubes and incubated at 37 °C in 5% CO_2_ for 2–3 h. Subsequently, 500 µL of the sample was centrifuged at 5000× *g* for 15 min in 0.22 µm filter tubes (Costar™, Spin-X™ Centrifuge Tube Filters CA Membrane) to remove bacteria and debris and stored at 4 °C until a new spot test was carried out.

When another confluent lysis zone was obtained, a double agar overlay assay was carried out to determine the concentration of infectious phage particles.

### 4.6. Control Experiments

To exclude the interference of spontaneous induction of prophages from the bacterial hosts, bacterial strains used for enrichment were cultured in GCB+ for 24 h at 37 °C in 5% CO_2_. Obtained cultures were either centrifuged at 5000× *g* for 15 min in 0.22 µL m filter tubes (Costar™, Spin-X™ Centrifuge Tube Filters CA Membrane) or treated with chloroform to lyse bacteria and tested for lysis zones using the spot test. In addition, pure ESwab™ medium was directly tested for lysis using the spot test to exclude bacterial killing by media components.

*E. coli* phage GEC-3S (The Eliava IBMV, Tbilisi, Georgia) [44] was used to test the efficiency of our phage isolation procedure and to assess phage stability in ESwab™ medium.

Hundred-fold dilutions of phage GEC-3S with initial titer of 10.1 log PFU/mL were made in ESwab™ medium, centrifuged at 5000× *g* for 15 min through a 0.22 µm filter, and stored at −80 °C for up to a one-year period. After storage, phage-ESwab™ samples were thawed and titered using the double-agar overlay assay. Minimal concentration of phage GEC-3S tested was 2 log PFU/mL. In brief, 1 mL of phage dilutions in ESwab™ medium was added to 100 µL of *E. coli* strain DSM 613 grown overnight on Blood Agar at 37 °C. The phage–bacteria mixture was vortexed briefly and added to 4 mL preheated GCB+ top agar (0.6% agar) and overlaid on GCB agar plates. Plates were overnight incubated at 37 °C until plaques could be detected and counted.

### 4.7. Ethics Approval

Ethics approval for this study was obtained from ITM’s Institutional Review Board (1276/18) and the Ethics Committee of the University of Antwerp (19/06/058).

## 5. Conclusions

This is the first report of a pilot screening of clinical samples for the presence of *N. gonorrhoeae* phages. A strictly virulent *N. gonorrhoeae* phage would be of considerable utility in the fight against extensively resistant *N. gonorrhoeae* infections.

Further studies may consider screening for phages at other anatomical sites (e.g., stool samples, urine) or in environmental settings (e.g., toilet sewage water of sex clubs or sexually transmitted infection clinics) where *N. gonorrhoeae* can be found. It may also be possible to engineer a related phage to target *N. gonorrhoeae* [45,46]. The phages that target *N. meningitidis* would be a logical place to start [19,21].

## Figures and Tables

**Figure 1 antibiotics-11-00268-f001:**
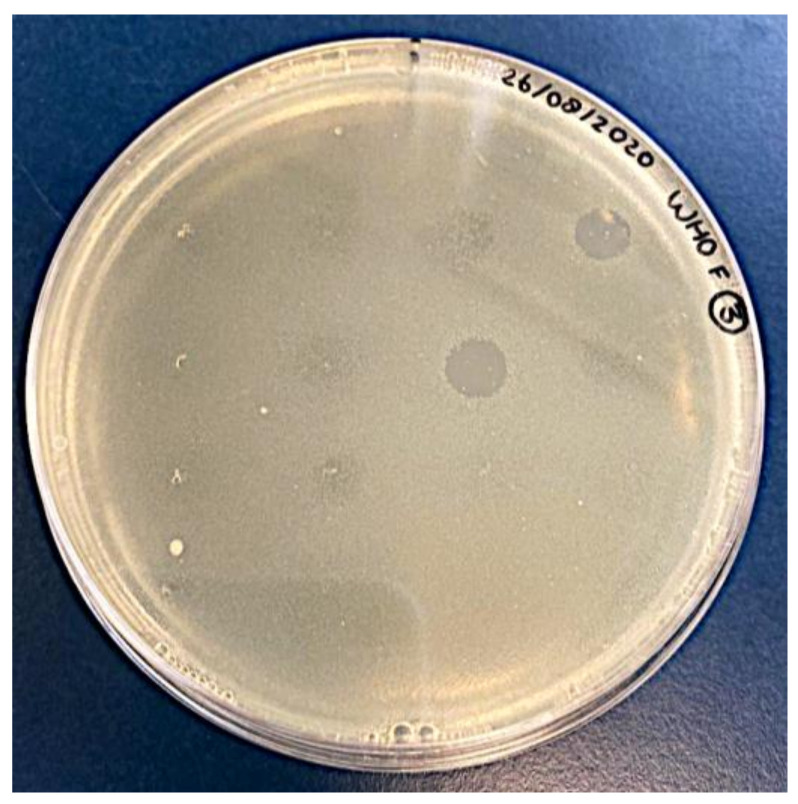
Typical spot test result: semi-confluent lysis zones obtained on *N. gonorrhoeae* reference strain WHO-F.

**Table 1 antibiotics-11-00268-t001:** Oropharyngeal and anorectal ESwabs™ collected from men who have sex with men from two clinical studies at ITM Antwerp, Belgium, used for *N. gonorrhoeae* phage screening.

Study	Condition	Oropharyngeal Swab	Anorectal Swab
Resistogenicity study	Pre antibiotic use	10	10
	Post antibiotic use	10	8
PReGo study	Baseline	64	/
	Post Listerine use	54	/
	Post placebo use	54	/

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
