# Peer review of "Screening of Anorectal and Oropharyngeal Samples Fails to Detect Bacteriophages Infecting Neisseria gonorrhoeae"

_antibiotics, 2022, doi:10.3390/antibiotics11020268_

Round 1
Reviewer 1 Report
A strictly virulent N. gonorrhoeae phage is very useful in the fight against extensively resistant N. gonorrhoeae infections. In this study, they screened for bacteriophages able to lyze N. gonorrhoeae in oropharyngeal and anorectal swabs of men. Unfortunately, no anti-gonococcal bacteriophage was found. Hence, it may be not suitable for publication in Antibiotics at this stage. As the authors mentioned, further studies are needed by screening for phages at other anatomical sites or in environmental settings.
Author Response
Dear Reviewer,
Thank you for the positive review. Although we did not find phages targeting Neisseria gonorrhoeae in anorectal and oropharyngeal samples, we think it is important to publish these results to prevent senseless repetitions. In addition, the methods described could be useful for future phage isolation studies in different samples.
Yours sincerely,
Jolein Laumen
On behalf of the authors
Institute of Tropical Medicine, Antwerp, Belgium
Department of Clinical Sciences, Unit of Sexually Transmittable Infections
Reviewer 2 Report
The authors here have done their due diligence to look for a phage for Neisseria gonorrhoeae in a highly probable niche. The subject matter of difficulty in isolating robustly lytic phage against Neisseria gonorrhoeae has puzzled the phage research community for a long time. Although this study confirms the virtual non-existence of lytic or temperate phage of growth capability among the large set of samples, it might motivate others either to find a natural phage or engineer one to target the clinical strains of Neisseria gonorrhoeae. On this merit and thus to reward their extensive work on the phage hunt, I recommend this information to be published.
Author Response
Dear Reviewer,
Thank you for the positive review.
Yours sincerely,
Jolein Laumen
On behalf of the authors
Institute of Tropical Medicine, Antwerp, Belgium
Department of Clinical Sciences, Unit of Sexually Transmittable Infections
Reviewer 3 Report
The short paper from Laumen et al investigates the presence of phages in gonococci isolated from MSM at the oropharnyges and anorectal sites. It is a study of 74 men with 210 swabs screened. Overall, the paper is well-written and controlled, especially for the techniques of phage isolation (by using the E.coli phage system as a control). I cannot argue with the conclusion that gonococci isolated from these body sites do not produce phages, despite the presence of prophages in the genomes.
Very minor comments:
- I would not call this a large-scale study. Please refer to it as a pilot study, as the number is still low.
- Why do the authors think that they may pick up phage from gonococci isolated from other sites/sources (Conclusions)? Can they provide a comment please in conclusions.
- Out of interest, what is the nature of the essential oil used in the study mentioned, and is this a limitation of the study?
Author Response
We have replied to the comments and questions raised by Reviewer 3.

Round 2
Reviewer 1 Report
No further improvement comments. Thanks.